# SuperCoder: Assembly Program Superoptimization with Large Language Models

## Abstract

Superoptimization is the task of transforming a program into a faster one while preserving its input–output behavior. In this work, we investigate whether large language models (LLMs) can serve as superoptimizers, generating assembly programs that outperform code already optimized by industry-standard compilers. We construct the first large-scale benchmark for this problem, consisting of 8,072 assembly programs averaging 130 lines, in contrast to prior datasets restricted to 2–15 straight-line, loop-free programs. We evaluate 23 LLMs on this benchmark and find that the strongest baseline, Claude-opus-4, achieves a 51.5% test-passing rate and a 1.43× average speedup over gcc -O3. To further enhance performance, we fine-tune models with reinforcement learning, optimizing a reward function that integrates correctness and performance speedup. Starting from Qwen2.5-Coder-7B-Instruct (61.4% correctness, 1.10× speedup), the fine-tuned model SuperCoder attains 95.0% correctness and 1.46× average speedup, with additional improvement enabled by Best-of-N sampling and iterative refinement. Our results demonstrate, for the first time, that LLMs can be applied as superoptimizers for assembly programs, establishing a foundation for future research in program performance optimization beyond compiler heuristics. Our code is available at https://anonymous.4open.science/r/SuperCoder/.

## 1 Introduction

Superoptimization is the task of transforming a program into a faster one while preserving its input-output behavior. In this work, we investigate whether large language models (LLMs) can perform superoptimization by generating assembly code that surpasses the performance of compiler outputs.

Decades of research have tackled the problem of code optimization, giving rise to two main approaches. The first develops better algorithms for rule-based transformations in compilers (Wolf & Lam, 1991). However, given the vast space of possible transformations, compiler-optimized code is not guaranteed to be optimal and often leaves performance untapped (Center et al., 1971; Theodoridis et al., 2022). The second, superoptimization, develops search algorithms that directly explore the space of all possible programs, aiming to discover the correct variant with the best performance rather than relying on a fixed set of transformation rules (Schkufza et al., 2013).

Superoptimization is an aggressive form of program optimization that can outperform compiler-optimized code, yet existing literature has focused on very short, straight-line assembly programs without loops. Prior work has primarily relied on CPU-based search heuristics, which fail to scale to larger programs (Schkufza et al., 2013; Phothilimthana et al., 2016; Koenig et al., 2021b); available datasets include at most 15 lines of straight-line assembly (Koenig et al., 2021a).

In this work, we explore the use of LLMs as a superoptimizer to improve the performance of assembly code. In contrast to most prior work on code generation from natural language (Chen et al., 2021; Austin et al., 2021; Hendrycks et al., 2021; Zhuo et al., 2024), we tackle a fundamentally different and more technically demanding task: improving assembly code that has already been optimized by the industry-standard compiler at its highest optimization level (gcc -O3). Compilers have been refined over decades of expert-driven development, and surpassing them remains a central challenge in programming languages, as compilers form the foundation of all software.

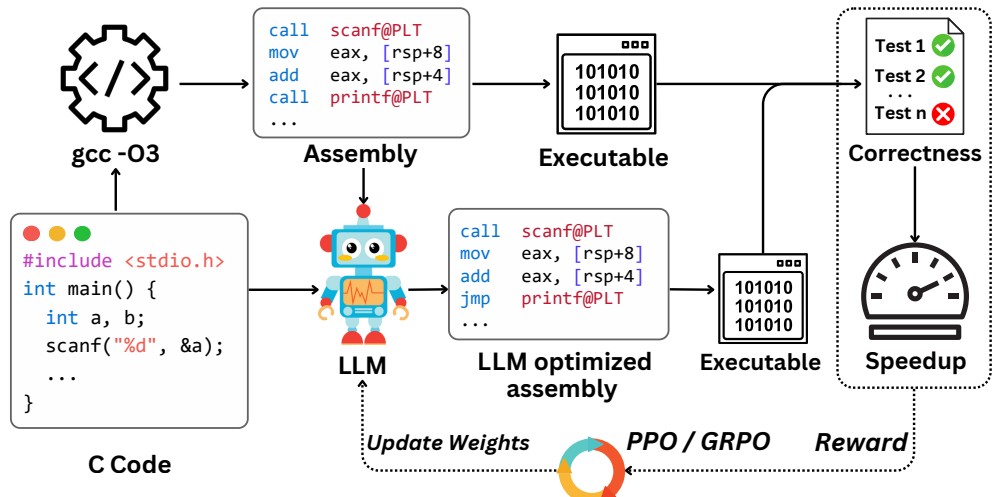

Figure 1: Overview of the assembly code optimization task. Given a C program and its baseline assembly from gcc -O3, an LLM is fine-tuned with PPO or GRPO to generate improved assembly. The reward function reflects correctness and performance based on test execution.

Unlike high-level programming languages (e.g., Python or C), large-scale, high-quality assembly datasets are scarce. As the first study in this direction, we construct a dataset of 8,072 assembly programs. Each instance includes input–output test cases and baseline assembly generated by the compiler at its highest optimization level (gcc -O3), which serves as the starting point for further optimization. In contrast, the datasets commonly used in the superoptimization community (Warren, 2013; Schkufza et al., 2013; Koenig et al., 2021a) are either extremely limited in size, containing only 25 programs, or consist of toy examples with 2 to 15 instructions without loops. Our dataset is substantially larger, with assembly programs averaging 130 lines and including loops. Our test suites achieve 96.2% line and 87.3% branch coverage, demonstrating strong test quality. Our dataset represents a substantial step forward in scale for evaluating superoptimization techniques.

Beyond evaluating existing models, we also apply reinforcement learning (RL) for fine-tuning to further enhance their capabilities. We use widely adopted algorithms, including Proximal Policy Optimization (PPO) and Group Relative Policy Optimization (GRPO), to train an LLM with a reward function that integrates both correctness and performance speedup. Prior work on LLM-based performance optimization has explored alternative methodologies such as supervised fine-tuning (Shypula et al., 2024), chain-of-thought prompting (Liu et al., 2024c), agent-based frameworks (Wei et al., 2025b;c), and preference learning (Du et al., 2024). Our approach optimizes speedup explicitly in the reward function, making reinforcement learning well suited to superoptimization. To our knowledge, this is the first application of reward-based RL to LLMs for code performance optimization, with correctness and speedup jointly encoded in the objective.

We evaluate 23 LLMs on this task and find that the best-performing model, Claude-opus-4, achieves a 51.5% test-passing rate and an average speedup of 1.43× over the compiler-optimized baseline (gcc -O3). Our reinforcement learning approach is highly effective: starting from the base model Qwen2.5-Coder-7B-Instruct, which achieves 61.4% correctness and a modest 1.10× speedup, the fine-tuned model SuperCoder attains 95.0% correctness and 1.46× average speedup, with further improvement enabled by Best-of-N sampling and iterative refinement.

In summary, our contributions are as follows:

- We are the first to introduce superoptimization as a task for LLMs, a technically demanding challenge that aims to improve assembly code already optimized by industry-standard compilers.

- We construct the first large-scale dataset of 8,072 assembly programs, averaging 130 lines. This far surpasses prior loop-free datasets under 15 lines and marks a substantial step forward in scale and realism for evaluating superoptimization techniques.

- We evaluate 23 LLMs on the benchmark and show that RL-based training substantially improves performance: fine-tuning Qwen2.5-Coder-7B-Instruct (61.4% correctness, 1.10× speedup) results in SuperCoder with 95.0% correctness and 1.46× speedup, with further gains enabled by Best-of-N sampling and iterative refinement.

## 2 RELATED WORK

**Large Language Models for Code.** Benchmarks for evaluating large language models (LLMs) on code generation from natural language specifications have received increasing attention. Notable examples include HumanEval (Chen et al., 2021), MBPP (Austin et al., 2021), APPS (Hendrycks et al., 2021), and more recent efforts (Liu et al., 2023b; Li et al., 2024; Xia et al., 2024b; Zhuo et al., 2024). In parallel, many models have been developed to enhance code generation capabilities, such as Codex (Chen et al., 2021), AlphaCode (Li et al., 2022), CodeGen (Nijkamp et al., 2022), InCoder (Fried et al., 2022), StarCoder (Li et al., 2023), DeepSeek-Coder (Guo et al., 2024), Code Llama (Roziere et al., 2023), and others (Hui et al., 2024; Wei et al., 2025e). Beyond code generation, LLMs have been applied to real-world software engineering tasks including automated program repair (Xia & Zhang, 2022; Xia et al., 2023b), software testing (Xia et al., 2023a; Deng et al., 2024), bug localization (Yang et al., 2024a), transpilation (Yang et al., 2024c; Bhatia et al., 2024), equivalence checking (Wei et al., 2025a), and synthesis (Wei et al., 2025d). SWE-bench (Jimenez et al., 2023) integrates these tasks into a benchmark for resolving real GitHub issues. Building on this, SWE-agent (Yang et al., 2024b) and subsequent works (Xia et al., 2024a; Wei et al., 2025f) employ an agent-based framework that leverages LLMs to improve the issue resolution process.

Recent work has also explored LLMs for improving program performance. CodeRosetta (Tehrani-Jamsaz et al., 2024) targets automatic parallelization, such as translating C++ to CUDA. Other efforts focus on optimizing Python code for efficiency (Du et al., 2024; Liu et al., 2024c) or enabling self-adaptation (Huang et al., 2024), and improving C++ performance (Shypula et al., 2024). Of particular relevance are approaches to low-level code optimization (Wei et al., 2024; Ouyang et al., 2025). The LLM Compiler foundation models (Cummins et al., 2024; 2025) are primarily designed for code size reduction and binary disassembly, whereas our work focuses on optimizing assembly code for performance. LLM-Vectorizer (Taneja et al., 2025) offers a formally verified solution for auto-vectorization, a specific compiler pass. In contrast, our work does not restrict the optimization type and uses test-case validation.

**Learning-Based Code Optimization.** The space of code optimization is vast, and many approaches have leveraged machine learning to improve program performance. A classic challenge in compilers is the phase-ordering problem, where performance depends heavily on the sequence of optimization passes. AutoPhase (Haj-Ali et al., 2020) uses deep reinforcement learning to tackle this, while Coreset (Liang et al., 2023) employs graph neural networks (GNNs) to guide optimization decisions. Modern compilers apply extensive rewrite rules but offer no guarantee of optimality. Superoptimization seeks the most efficient program among all semantically equivalent variants of the compiler output. Traditional methods use stochastic search, such as MCMC sampling (Schkufza et al., 2013), with follow-up work improving scalability (Phothilimthana et al., 2016; Bunel et al., 2016) and extending to broader domains (Sharma et al., 2015; Churchill et al., 2017). These rely on formal verification for correctness, restricting them to small, loop-free programs. In contrast, our approach uses test-based validation, enabling optimization of general programs with loops. With the rise of deep learning, substantial attention has turned to optimizing GPU kernel code. AutoTVM (Chen et al., 2018) pioneered statistical cost model-based search for CUDA code optimization, followed by methods such as Ansor (Zheng et al., 2020), AMOS (Zheng et al., 2022), and others (Shao et al., 2022; Zhao et al., 2024; Wu et al., 2024).

More recently, LLMs have been explored as code optimizers (Shypula et al., 2024; Grubisic et al., 2024; Wei et al., 2024; 2025b;c), with growing interest in reinforcement learning that guides generation through reward signals (Dou et al., 2024; Wei et al., 2025f). Rewards are often derived from unit-test correctness (Le et al., 2022; Shojaee et al., 2023; Liu et al., 2023a) or binary preference signals (Liu et al., 2024b; Du et al., 2024). To our knowledge, this is the first work to apply reinforcement learning to optimize code performance with LLMs, with concurrent efforts exploring CUDA kernel optimization (Li et al., 2025; Baronio et al., 2025).

## 3 METHODOLOGY

### 3.1 TASK DEFINITION

Let $C$ be a program written in a high-level language such as C. A modern compiler like gcc can compile $C$ into an x86-64 assembly program $P = gcc(C)$, which can then be further assembled into an executable binary. The assembly program $P$ serves as an intermediate representation that exposes low-level optimization opportunities, making it suitable for aggressive performance improvement. We assume the semantics-preserving nature of the compilation process, i.e., $[\![C]\!] = [\![P]\!]$, so that the behavior of the assembly program $P$ is identical to that of the source program $C$.

In theory, the goal is to produce a program $P'$ that is functionally equivalent to $P$ across the entire input space $\mathcal{X}$, i.e., $P(x) = P'(x)$ for all $x \in \mathcal{X}$. Since verifying this property is undecidable in general, we approximate equivalence using a finite test set $\mathcal{T} = \{(x_i, y_i)\}_{i=1}^n$, where each input-output pair $(x_i, y_i)$ captures the expected behavior of $C$.

We say that an assembly program $P'$ is *valid* if it can be successfully assembled and linked into an executable binary. Let $\texttt{valid}(P') \in \{\texttt{True}, \texttt{False}\}$ denote this property. Based on all that we said above, we define the set of *correct* programs as:

$$\mathcal{S}(P) = \{P' \mid \texttt{valid}(P') \ \wedge \ \forall (x_i, y_i) \in \mathcal{T}, \ P'(x_i) = y_i\}.$$

**Performance and Speedup.** Let $t(P)$ denote the execution time of $P$ on the test set $\mathcal{T}$, and let $t(P')$ be the corresponding execution time for $P'$. The speedup of $P'$ relative to $P$ is defined as follows. If the LLM-generated program is invalid or slower, we fall back to the baseline and assign a speedup of 1.

$$\text{Speedup}(P') = \begin{cases} \frac{t(P)}{t(P')} & \text{if } P' \in \mathcal{S}(P) \text{ and } t(P') < t(P), \\ 1 & \text{otherwise.} \end{cases}$$

**Optimization Objective.** The objective is to generate a candidate program $P'$ that maximizes Speedup$(P')$. Only programs in $\mathcal{S}(P)$ are eligible for speedup; any candidate that fails to compile into a binary or produces incorrect outputs is assigned a default speedup of 1. This reflects a practical fallback: when the generated program is invalid, the system can revert to the baseline $P$, compiled with gcc -O3, which defines the 1× reference performance. Although $\mathcal{S}(P)$ captures the correctness criteria, we do not restrict the LLM to generate only valid programs. Instead, the model produces arbitrary assembly code, and correctness is validated post hoc via compilation and test execution. We train an LLM using reinforcement learning (see Section 3.3) to generate candidates that both satisfy correctness and achieve performance improvements.

### 3.2 DATASET CONSTRUCTION

We construct our dataset using C programs from CodeNet (Puri et al., 2021), a large-scale corpus of competitive programming submissions. CodeNet is a well-established and widely used benchmark in the AI-for-code community (Li et al., 2022; Shypula et al., 2024). Each dataset instance is a tuple $(C, P, \mathcal{T})$, where $C$ is the original C source code, $P = gcc(C)$ is the corresponding x86-64 assembly generated by compiling $C$ with gcc at the -O3 optimization level, and $\mathcal{T} = \{(x_i, y_i)\}_{i=1}^n$ is the test set. Since not all CodeNet problems include test inputs, we adopt those provided by prior work (Li et al., 2022) to define $x_i$, but discard their output labels. Instead, we regenerate each output $y_i$ by executing the original submission on input $x_i$, as many CodeNet programs are not accepted solutions, and even accepted ones do not reliably pass all test cases.

Given the scale of CodeNet, which contains over 8 million C and C++ submissions, we sample a subset for this study. To focus on performance-critical cases, we sample programs that exhibit the highest relative speedup from gcc -O0 (no optimization) to gcc -O3 (maximum optimization). Such a strategy serves two purposes: (1) it favors programs with complex logic that lead to suboptimal performance under -O0 and can be effectively optimized by -O3, and (2) it creates a more challenging setting by starting from code that has already benefited from aggressive compiler optimizations. The final dataset consists of 7,872 training programs and 200 held-out evaluation programs, with

additional statistics provided in Section 4. We also evaluate our method on an unbiased, randomly sampled dataset distribution in Appendix A.5.

## 3.3 Reinforcement Learning

We conceptualize our task as a standard contextual multi-armed bandit problem (Lu et al., 2010), defined by a context space $\mathcal{S}$, an action space $\mathcal{A}$, and a reward function $r : \mathcal{S} \times \mathcal{A} \to \mathbb{R}$. Each context $s \in \mathcal{S}$ represents a problem instance, comprising the source program $C$, its baseline assembly $P$, and the associated test cases $\mathcal{T}$. An action $a \in \mathcal{A}$ corresponds to generating a candidate assembly program $\tilde{P}$. The reward function $r(s, a)$ evaluates the quality of the generated program based on correctness and performance. We will describe different designs of the reward function later. A policy $\pi : \mathcal{S} \to \Delta(\mathcal{A})$ maps a context $s$ to a probability distribution over actions and samples an action $a \in \mathcal{A}$ stochastically. Given a distribution $\mu$ over problem instances, the expected performance of a policy $\pi$ under reward function $r$ is expressed as $\mathbb{E}_{s \sim \mu, a \sim \pi(\cdot|s)} [r(s, a)]$. The objective is to find a policy that maximizes this expected reward.

**Optimization with PPO and GRPO.** We train the policy using two policy-gradient algorithms: *Proximal Policy Optimization* (PPO) (Schulman et al., 2017) and *Group Relative Policy Optimization* (GRPO) (Shao et al., 2024). PPO stabilizes training by constraining each update to remain close to the previous policy. It maximizes a clipped surrogate objective of the form $\mathbb{E}_{s,a} \left[ \min \left( \rho(\theta)\hat{A}, \ \text{clip}(\rho(\theta), 1 - \epsilon, 1 + \epsilon) \hat{A} \right) \right]$, where $\rho(\theta) = \pi_\theta(a \mid s)/\pi_{\theta_{\text{old}}}(a \mid s)$, $\hat{A}$ is the estimated advantage, and $\epsilon$ controls the clipping range. GRPO, in contrast, compares rewards among a group of sampled outputs and assigns a higher likelihood to relatively stronger ones, effectively normalizing advantages without requiring a value function. In both algorithms, rewards are based on the correctness and execution time of the generated program, eliminating the need for a separate reward model.

**Reward Function Design.** As defined in our contextual bandit setup, the reward function $r : \mathcal{S} \times \mathcal{A} \to \mathbb{R}$ assigns a scalar score to each (context, action) pair. Each context $s \in \mathcal{S}$ consists of the source program $C$, the baseline assembly $P$, and a test set $\mathcal{T} = \{(x_i, y_i)\}_{i=1}^n$. An action $a \in \mathcal{A}$ corresponds to a generation procedure that produces a candidate assembly program $\tilde{P} = \texttt{gen}(a)$.

We define two auxiliary metrics for computing reward:

$$\text{pass}(s, a) = \frac{1}{|\mathcal{T}|} \sum_{(x,y) \in \mathcal{T}} \mathbf{1}[\tilde{P}(x) = y], \quad \text{speedup}(s, a) = t(P)/t(\tilde{P}),$$

which denote the fraction of test cases passed and the speedup of the generated program $\tilde{P}$ relative to the baseline $P$. We use the following reward function during training:

$$r(s, a) = \begin{cases} 0, & \text{if } \text{pass}(s, a) < 1, \\ \text{speedup}(s, a), & \text{if } \text{pass}(s, a) = 1. \end{cases}$$

If a generated program fails to compile or does not pass all tests, its reward is set to 0, with no partial credit for partial correctness. Only when the code compiles and passes all tests is a positive reward assigned, equal to the achieved speedup.

## 3.4 Best-of-N Sampling, Supervised Fine-Tuning, and Iterative Refinement

**Best-of-N Sampling.** Generating multiple candidate programs and selecting the strongest one is a well-established strategy for improving code generation quality (Li et al., 2022; Ehrlich et al., 2025). In our setting, the best candidate refers to the program that is correct while achieving the fastest execution time. Best-of-N sampling is an inference-time technique that can boost performance, but it incurs additional cost because each candidate must be tested at runtime.

**Supervised Fine-Tuning.** To obtain training targets for supervised fine-tuning, we require reference solutions to the superoptimization task. However, superoptimization is inherently open-ended: beyond

the compiler baseline, there is no unique ground-truth program, and multiple distinct solutions may exist. We therefore apply best-of-8 sampling with the base model over the full training set and treat the highest-quality candidate for each instance as the ground truth. We then fine-tune the model using LoRA (Hu et al., 2022).

**Iterative Refinement.** Iterative refinement is a complementary inference-time technique that can be applied to any model to further improve its outputs. After each trial, we feed back the model's previous attempt: if the generated program fails to compile or fails any test cases, we include the corresponding compiler errors or test failures in the next prompt; if the model produces a correct program, we also include that successful attempt as part of the prompt.

## 4 EXPERIMENTAL SETUP

**Dataset.** We describe our dataset construction approach in Section 3.2. Each instance consists of a C source program $C$, the corresponding gcc -O3 compiled assembly $P$, and a set of test cases $\mathcal{T}$ for correctness evaluation. The final dataset contains 7,872 training programs and 200 evaluation programs, with average program lengths and test case counts summarized in Table 1, and additional analysis below.

| Split | # Prog. | Avg. Tests | Avg. LOC | |
|---|---|---|---|---|
| | | | C | Assembly |
| Training | 7,872 | 8.86 | 22.3 | 130.3 |
| Evaluation | 200 | 8.92 | 21.9 | 133.3 |

Table 1: Dataset statistics across training and evaluation splits. LOC = lines of code.

**Test Coverage.** Our dataset includes test cases for every program. Rather than relying directly on the original submissions, we re-run each program on its inputs to generate correct outputs, thereby fixing errors in prior datasets. The resulting test suites of our evaluation dataset achieve an average of 96.2% line coverage and 87.3% branch coverage, demonstrating high test quality.

**Speedup by Compilers.** We quantify compiler optimizations by comparing gcc -O0 with gcc -O3 on the evaluation dataset and observe a mean speedup of 2.65×. This demonstrates the substantial effect of compiler optimizations and confirms that performance improvements in our dataset are measurable. Building on this baseline, we investigate whether LLMs can further enhance performance beyond the 2.65× speedup provided by the compiler.

**Prompts.** For each instance, we construct a prompt that includes the original C program along with the generated assembly using gcc -O3. Test cases are withheld from the model. The model is instructed to generate the optimized assembly code. We show the prompt template in Appendix A.3.

**Metrics.** We evaluate each model using both correctness and performance metrics. *Compile pass* is the percentage of problems for which the generated assembly compiles to binary executable successfully, while *test pass* is the percentage of problems where the compiled code passes all test cases. For a given problem, any single failed test case is considered a failure for the test pass metric. Both metrics are computed across the entire validation set. For performance, we measure the relative speedup over the gcc -O3 baseline. As defined in Section 3.1, we assign a default speedup of 1× to any candidate that fails to compile, fails any test case, or is slower than the baseline. This reflects the practical setting where a system can fall back to the gcc -O3 output, resulting in no performance gain. We report the *25th*, *50th* (median), and *75th* percentiles of speedup to capture distributional behavior, along with the *average speedup* over the entire evaluation set.

**Models.** We evaluate 23 state-of-the-art language models spanning a diverse range of architectures. Our benchmark includes frontier proprietary models such as gpt-4o (Achiam et al., 2023), o4-mini, gemini-2.0-flash-001 (Team et al., 2023), and claude-3.7-sonnet, as well as open-source families such as Llama (Touvron et al., 2023), DeepSeek (Liu et al., 2024a), and Qwen (Hui et al., 2024). In addition, we include models distilled from DeepSeek-R1 (Guo et al., 2025) based on Qwen and Llama. Finally, we evaluate recent compiler-oriented foundation models (Cummins et al., 2024;

| Model | Compile Pass | Test Pass | Speedup Percentiles | | | Average Speedup |
|---|---|---|---|---|---|---|
| | | | 25th | 50th | 75th | |
| DS-R1-Distill-Qwen-1.5B | 0.0% | 0.0% | 1.00× | 1.00× | 1.00× | 1.00× |
| DeepSeek-R1 | 0.0% | 0.0% | 1.00× | 1.00× | 1.00× | 1.00× |
| DS-R1-Distill-Llama-70B | 5.5% | 0.0% | 1.00× | 1.00× | 1.00× | 1.00× |
| DS-R1-Distill-Qwen-14B | 11.5% | 0.5% | 1.00× | 1.00× | 1.00× | 1.00× |
| gpt-4o-mini | 44.5% | 1.0% | 1.00× | 1.00× | 1.00× | 1.00× |
| Llama-4-Maverick-17B | 77.5% | 7.0% | 1.00× | 1.00× | 1.00× | 1.02× |
| Llama-3.2-11B | 84.0% | 21.0% | 1.00× | 1.00× | 1.00× | 1.02× |
| gpt-4o | 81.0% | 5.0% | 1.00× | 1.00× | 1.00× | 1.02× |
| Llama-4-Scout-17B | 68.5% | 5.5% | 1.00× | 1.00× | 1.00× | 1.02× |
| o4-mini | 25.0% | 4.5% | 1.00× | 1.00× | 1.00× | 1.02× |
| gemini-2.0-flash-001 | 57.5% | 4.0% | 1.00× | 1.00× | 1.00× | 1.03× |
| Qwen2.5-72B | 59.5% | 7.5% | 1.00× | 1.00× | 1.00× | 1.03× |
| Llama-3.2-90B | 82.5% | 15.0% | 1.00× | 1.00× | 1.00× | 1.05× |
| Qwen2.5-Coder-7B | 77.9% | 61.4% | 1.00× | 1.00× | 1.00× | 1.10× |
| gpt-5 | 78.5% | 6.0% | 1.00× | 1.00× | 1.00× | 1.13× |
| DeepSeek-V3 | 94.0% | 43.0% | 1.00× | 1.00× | 1.40× | 1.21× |
| claude-3.7-sonnet | 94.5% | 58.5% | 1.00× | 1.10× | 1.45× | 1.22× |
| claude-sonnet-4 | 87.0% | 37.0% | 1.00× | 1.00× | 1.95× | 1.30× |
| claude-opus-4 | 90.0% | 51.5% | 1.00× | 1.58× | 2.03× | 1.43× |
| llm-compiler-7b-ftd | 2.0% | 2.0% | 1.00× | 1.00× | 1.00× | 1.00× |
| llm-compiler-13b-ftd | 2.5% | 2.0% | 1.00× | 1.00× | 1.00× | 1.01× |
| llm-compiler-7b | 55.0% | 54.0% | 1.00× | 1.00× | 1.00× | 1.09× |
| llm-compiler-13b | 60.5% | 59.5% | 1.00× | 1.27× | 1.63× | 1.34× |

Table 2: Comparison of LLMs on our assembly optimization benchmark. We report compilation success rate, test pass rate, and average speedup over the gcc -O3 baseline. All open-source models are instruction-tuned.

2025), pretrained on assembly code and derived from Code Llama, with a design focus on compiler tasks (listed as llm-compiler in Table 2). All open-source models are instruction-tuned.

**Performance Measurement.** To ensure an accurate performance evaluation, we use hyperfine (hyp, 2025), a benchmarking tool that reduces measurement noise by performing warmup runs followed by repeated timed executions. For each program's execution, we discard the first three runs and report the average runtime over the next ten runs.

**Implementation.** We implement our customized reinforcement learning reward functions within the VERL framework (Sheng et al., 2024), which enables fine-tuning of LLMs using PPO and GRPO. As part of this setup, we build a task-specific environment that handles program compilation, test execution, and runtime measurement, as detailed in Section 3.3. This environment provides the model with direct scalar feedback based on both functional correctness and execution performance.

**Training Configurations.** Among all evaluated models (see Table 2), we select Qwen2.5-Coder-7B-Instruct for training due to its strong correctness results and substantial room for performance improvement, while intentionally avoiding compiler-specific foundation models to preserve generality. Training is performed on a single node with four A100 GPUs. Full hyperparameter settings are provided in Appendix A.2.

## 5 RESULTS

### 5.1 EVALUATION OF DIFFERENT MODELS

**Main Results.** Table 2 reports results across evaluated models. Most perform poorly on this task, with only a few demonstrating effectiveness as superoptimizers. Most models struggle to generate performant assembly: the majority yield only 1.00× speedup, with low compile and test pass rates. Among all models, claude-opus-4 and claude-sonnet-4 perform best, achieving average speedups

| Model | Compile Pass | Test Pass | Average Speedup |
|---|---|---|---|
| Qwen2.5-Coder-7B (Base) | $77.9 \pm 0.8\%$ | $61.4 \pm 0.5\%$ | $1.10 \pm 0.01\times$ |
| SuperCoder (GRPO) | $95.0 \pm 0.0\%$ | $94.7 \pm 0.6\%$ | $1.44 \pm 0.07\times$ |
| SuperCoder (PPO) | $96.0 \pm 0.0\%$ | $95.0 \pm 0.0\%$ | $1.46 \pm 0.12\times$ |
| SuperCoder (Supervised fine-tuning) | $95.5 \pm 0.0\%$ | $92.5 \pm 0.0\%$ | $1.39 \pm 0.05\times$ |

Table 3: Performance of the base model and the models trained with RL or supervised fine-tuning. Results include compilation success, test pass rates, and average speedup, reported over 5 runs with 95% confidence intervals.

of 1.43× and 1.30×, respectively. Compiler foundation models (prefixed with llm-compiler-) are pretrained on assembly code and compiler IRs. Among them, llm-compiler-13b achieves a notable 1.34× speedup, whereas the fine-tuned variants (-ftd) perform poorly, likely because they were adapted for different tasks such as disassembling x86-64 into LLVM IR. These results suggest that while superoptimization is inherently difficult, some LLMs can be effective superoptimizers.

**Failure Modes.** Interestingly, models that are expected to achieve strong results perform (e.g., DeepSeek-R1, GPT-4o) perform poorly on the task of superoptimization, motivating our analysis of their failure modes. We find that DeepSeek-R1 consistently fails to generate valid assembly code, resulting in a 0% compilation rate. DeepSeek-R1 often produces verbose analysis instead of executable code, spending its entire output length on reasoning about instruction semantics and potential optimizations without actually implementing them.

We further analyze the failure modes of GPT-4o, which achieves a high compilation rate (81.0%) but exhibits poor correctness (only 5.0% test pass rate). The primary correctness issues are as follows: (1) missing critical directives and safety setup, such as stack canary initialization and `.cfi_*` metadata, which often lead to runtime crashes; (2) incorrect function call conventions, where repeated system calls like `scanf` are made without proper argument setup, causing undefined behavior; (3) semantic errors in core computations, including incorrect pointer usage or altered algorithm logic, which produce wrong outputs even when the code runs; and (4) over-simplified stack or register management, resulting in memory errors or invalid control flow. In summary, GPT-4o tends to sacrifice correctness in pursuit of optimization: it generates syntactically valid assembly but frequently violates low-level conventions necessary for correct and reliable execution.

## 5.2 Effectiveness of RL Training

**Improvement.** Table 3 presents the results of RL training, averaged over 5 runs with 95% confidence intervals to provide more statistical confidence in the reported improvements. We select Qwen2.5-Coder-7B-Instruct for RL training due to its strong test pass rate (61.4%) among models. After RL training with PPO, the fine-tuned model SuperCoder attains 95.0% correctness and improves average speedup from 1.10× to 1.46×. Its speedup percentiles are 1.17 ± 0.03× (25th), 1.35 ± 0.04× (50th), and 1.64 ± 0.08× (75th) respectively, outperforming the majority of evaluated models.

**PPO versus GRPO.** We evaluate both PPO-trained and GRPO-trained models and find their performance to be nearly identical. SuperCoder trained with GRPO attains 94.7 ± 0.6% correctness and 1.44 ± 0.07× average speedup, which is comparable to SuperCoder trained with PPO (95.0 ± 0.0% correctness and 1.46 ± 0.12× average speedup).

## 5.3 Results from Supervised Fine-Tuning and Inference-Time Methods

**Best-of-N Sampling.** We evaluate best-of-N sampling for three models: claude-opus-4 (the strongest baseline in Table 2), Qwen2.5-Coder-7B (base), and SuperCoder (our PPO-trained model). Results are shown in Figure 2. Notably, the base model's best-of-8 speedup is close to the PPO-trained model's best-of-1 result, and the RL-trained model itself can still be improved with best-of-N sampling (i.e., from 1.46× in the single-sample setting to 1.93× with best-of-8 sampling).

**Supervised Fine-Tuning.** We describe our supervised fine-tuning approach in Section 3.4. Table 2 reports results averaged over five runs with 95% confidence intervals. While supervised fine-tuning

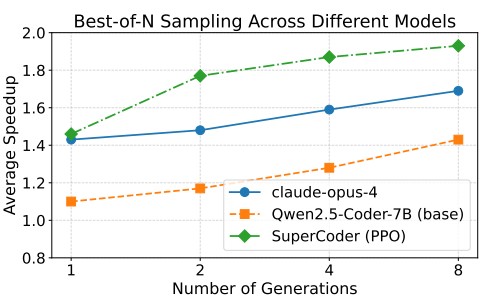

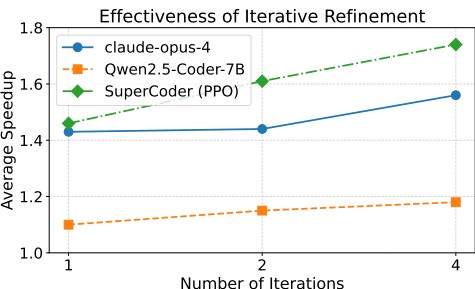

Figure 2: Best-of-N sampling results.   Figure 3: Iterative refinement results.

improves performance, RL achieves slightly stronger results. We believe that RL is a natural fit for the open-ended nature of superoptimization, as RL directly optimizes for correctness and speedup rather than imitating existing examples.

**Iterative Refinement.**   Figure 3 shows the results of iterative refinement, where the model receives feedback about compilation failures, test failures, or performance for self-reflection. All three models exhibit improvements as the number of refinement iterations increases, with the effect being most pronounced for our RL fine-tuned model.

### 5.4 ANALYSIS OF LEARNED PROGRAM TRANSFORMATIONS

To better understand why LLMs can further optimize assembly programs already optimized by industry-standard compilers, we analyze all 200 evaluation programs by comparing the gcc -O3 output with the assembly generated by our PPO-trained model SuperCoder. We categorize the learned transformations into four types: (1) code layout and instruction scheduling, which account for 98.5% of the changes and include basic block placement, instruction alignment, and reordering to improve cache behavior and hide latencies; (2) register allocation, representing 16% of the changes, where the model selects alternative registers; (3) control flow optimization (1.5%); and (4) instruction selection (1.0%). Since a single problem's transformation may belong to multiple categories, the percentages do not sum to 100%. Unfortunately, there is currently no automated tool that can further reliably explain performance speedups at the assembly level. Automated analysis and explanation of performance differences in assembly code remains a challenging and unsolved problem.

The observed gains and our analysis suggest that LLMs can produce reasonable code transformations that meaningfully improve assembly performance. These results indicate that LLMs can uncover nuanced optimization opportunities beyond the reach of traditional compiler heuristics, suggesting that compilers themselves still have room for improvement.

### 5.5 COMPARISON WITH SIMPLE RANDOM PERTURBATIONS

To assess whether the improvements achieved by LLMs stem from systematic strategies learned during training rather than simple random variation in assembly programs, we compare against a baseline that applies simple random perturbations to assembly code. While such perturbations do not necessarily improve performance, running them multiple times and selecting the fastest version allows us to compute a speedup. These perturbations preserve correctness but can still influence performance, and we report speedup using the minimum observed runtime across 10 runs.

To randomly perturb the assembly, we compile the binaries with options that randomize both the stack and program load addresses. This alters code layout in memory, which can affect performance but not correctness. Speedup is measured as the ratio of the fastest run to the original runtime over 10 randomized runs. Repeating the experiment five times yields a speedup of 1.19 ± 0.01× (95% CI), significantly lower than 1.46 ± 0.12× achieved by SuperCoder. This indicates that SuperCoder systematically exploits performance opportunities rather than relying on simple random perturbations, particularly since it produces each result in a single shot without retries or iterative refinement.

## 6 DISCUSSION

**Prompt Optimization Methods.** We experimented with few-shot in-context learning and found that adding more examples does not reliably improve performance and often degrades it, consistent with prior observations in code optimization (Shypula et al., 2024). We also evaluated GEPA (Agrawal et al., 2025), an evolutionary prompting framework that uses natural language reflection to derive optimization rules, but observed only minimal gains. Additional results are provided in Appendix A.4.

**Alternative Reward Function Design.** Besides the reward function presented in Section 3.3, we also evaluate an alternative design. The original design assigns zero reward whenever any test fails. In contrast, the alternative assigns (i) a reward of $-1$ if the program fails to compile, (ii) a partial reward equal to the fraction of passed tests if only some tests succeed, and (iii) a reward of $1 + \alpha \cdot$ speedup once all tests pass. Training the base model with this design yields an average speedup of 1.38×. Varying the scaling factor (5 or 10) has a negligible effect, and the result remains worse than the 1.46× achieved by SuperCoder with the original reward. This suggests that directly optimizing for the terminal speedup reward is more effective.

**Direct Compilation from C.** We also examine a more challenging setting where gcc -O0 assembly is not provided. Instead, LLMs receive only the C source code and are asked to generate assembly directly. This setup leads to a sharp drop in performance: for example, SuperCoder, which attains 95.0% correctness with the assembly baseline, fails to produce any compilable code without it. Similar degradation occurs for other models such as llm-compiler-13b and Claude models. These results indicate that, at least in their current state, LLMs can act as superoptimizers building on compiler outputs, but cannot replace compilers themselves.

**Case Study.** We illustrate in Appendix A.1 a representative example where an LLM discovers an optimization beyond the reach of a state-of-the-art compiler. The original C function computes the population count (i.e., the number of set bits) by repeatedly shifting the input and accumulating its least significant bit. The assembly produced by gcc -O3 preserves this loop structure, relying on explicit bitwise operations and conditional branches. In contrast, Claude-opus-4 produces an efficient implementation that replaces the entire loop with a single `popcnt` instruction, performing the same computation in one operation and reducing both instruction count and runtime overhead.

**Limitation and Potential Directions.** A limitation of our approach is the lack of formal correctness guarantees. The absence of a general formal equivalence checker is a well-known theoretical limitation, grounded in the undecidability of program equivalence. Our evaluation relies on test-based validation, which is common in with prior work (Shypula et al., 2024; Du et al., 2024). Moreover, our evaluation achieves 96.2% line coverage, which substantially reduces the risk of undetected errors. Encouragingly, researchers are already building formal verification tools for GPU kernels (Dubey et al., 2025), inspired in part by advances in LLM-generated GPU code. We hope that our work motivates the development of more general and scalable assembly verification tools.

Compared with traditional compiler optimization, reinforcement learning introduces substantial computational overhead during training. This cost, however, is incurred only once: once trained, the model can generate optimized assembly efficiently with low latency.

In addition, most superoptimization research has centered on x86-64, and our work follows this focus. Extending the methodology to other targets such as ARM, RISC-V, or GPU kernels could greatly broaden its applicability and impact.

## 7 CONCLUSION

We investigated whether LLMs can act as superoptimizers, generating assembly programs that outperform code already optimized by industry-standard compilers. To support this study, we introduced the first large-scale benchmark of 8,072 assembly programs. Evaluating 23 models revealed the difficulty of the task, with most failing to achieve meaningful gains. By fine-tuning with reinforcement learning, our model SuperCoder improved both correctness and performance, reaching 95.0% test pass rate and an average speedup of 1.46× over gcc -O3. We also show that Best-of-N sampling and iterative refinement can bring additional improvement. These results demonstrate, for the first time, that LLMs can be applied as superoptimizers for assembly code, opening new opportunities for performance optimization beyond compiler heuristics.

## LLM USAGE

Large Language Models (LLMs) are the primary subject of study in this paper. In addition, we used LLMs as a general-purpose writing assistant to polish the presentation and improve readability.

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

# A APPENDIX

## A.1 CASE STUDY

| **C Code** | **GCC -O3 Output** | **Claude-Opus-4** |

```c
int f(unsigned long x)
{
  int res = 0;
  while (x > 0)
  {
    res += x & 1;
    x >>= 1;
  }
  return res;
}
```

```asm
.L0:
 xorl %eax, %eax
 testq %rdi, %rdi
 je .L2
.L1:
 movq %rdi, %rdx
 andl $0x1, %edx
 addq %rdx, %rax
 shrq $0x1, %rdi
 jne .L1
 retq
.L2:
 retq
```

```asm
.L0:
 popcnt %rdi, %rax
 retq
```

Figure A1: Case study comparing the C code, baseline assembly produced by gcc -O3, and optimized assembly generated by Claude-Opus-4. The model successfully replaces the loop with the specialized hardware instruction popcnt, resulting in a significantly more concise implementation.

## A.2 TRAINING CONFIGURATIONS

| Component | Setting |
|---|---|
| Base model | Qwen2.5-Coder-7B-Instruct |
| Actor's learning rate | 1e-6 |
| Critic's learning rate | 1e-5 |
| Batch size | 16 |
| Epoch | 1 |
| Max prompt length | 2000 tokens |
| Max response length | 2000 tokens |
| Gradient checkpointing | Enabled (both actor and critic) |
| Rollout temperature | 0.5 |
| Hardware | 4× A100 GPUs |

Table A1: Key training configurations for PPO fine-tuning.

## A.3 PROMPT TEMPLATE

```
Prompt Template

Given the following C code and assembly code, your task is to
generate highly optimized x86-64 assembly code.

C Code:

<C code here>

Assembly Code:

<baseline assembly code here produced by gcc -O3>

Only output the optimized assembly code.  Do not include any other
text.  Do not write any comments in the assembly code.  Wrap the
assembly code in assembly tags.
Optimized Assembly Code:
```

| Model | Shots | Compile Pass (%) | Test Pass (%) | Avg. Speedup |
|---|---|---|---|---|
| claude-opus-4 | 0-shot | 90.0 | 51.5 | 1.43× |
| claude-opus-4 | 2-shot | 95.0 | 15.0 | 1.13× |
| claude-opus-4 | 4-shot | 95.0 | 12.5 | 1.11× |
| SuperCoder (PPO) | 0-shot | 96.0 | 95.0 | 1.46× |
| SuperCoder (PPO) | 2-shot | 94.0 | 90.5 | 1.59× |
| SuperCoder (PPO) | 4-shot | 93.0 | 81.0 | 1.54× |
| Qwen2.5-Coder-7B (Base) | 0-shot | 77.9 | 61.4 | 1.10× |
| Qwen2.5-Coder-7B (Base) | 2-shot | 70.5 | 35.0 | 1.10× |
| Qwen2.5-Coder-7B (Base) | 4-shot | 80.5 | 30.5 | 1.06× |

Table A2: Comparison of 0-shot, 2-shot, and 4-shot prompting across different models.

### A.4 PROMPT OPTIMIZATION METHODS

**Few-shot Prompting.** We evaluate 0-shot, 2-shot, and 4-shot prompting across three models. Table A2 shows that adding more in-context examples does not reliably improve performance and often degrades it. This observation is consistent with prior work (Shypula et al., 2024), which finds that few-shot examples can bias the model and lead to incorrect task understanding.

**Prompt Evolution Framework.** We experimented with GEPA (Agrawal et al., 2025), an evolutionary prompting framework that uses natural language reflection to learn high-level optimization rules from trial and error. We used gpt-4o as the model under evaluation and gpt-5 as the reflection model. GEPA yielded only modest gains: compilation pass increased from 81.0% to 84.0% and test pass from 5.0% to 7.5%, while performance speedup remained essentially unchanged. We suspect this is because assembly optimization requires substantial domain knowledge that is difficult to capture by modifying the prompt alone.

### A.5 ADDITIONAL EVALUATION ON RANDOMLY SAMPLED PROGRAMS

Our main dataset samples programs with larger speedup from -O0 to -O3 speedup. To evaluate how much dataset distribution affects our method, we conduct an additional experiment on 200 programs *randomly sampled* from CodeNet, and we ensure that this new set is fully disjoint from both our training and evaluation splits and *not* selected based on compiler speedup characteristics. Across these unbiased sampled programs, the conclusions remain largely unchanged: the PPO-trained model consistently produces correct and faster assembly code. This indicates that the improvements learned through reinforcement learning generalize beyond the specific distribution used for training and primary evaluation.

| Model | Compile Pass (%) | Test Pass (%) | Avg. Speedup |
|---|---|---|---|
| Qwen2.5-Coder-7B | 66.5 | 40.0 | 1.07× |
| claude-opus-4 | 94.0 | 43.0 | 1.10× |
| **SuperCoder (PPO)** | **98.5** | **93.5** | **1.57×** |

Table A3: Performance on 200 randomly sampled programs from CodeNet, disjoint from all training and evaluation splits. Unlike the main dataset, these programs are *not* selected for large gcc -O0 to gcc -O3 speedup. The PPO-trained model maintains strong correctness and performance gains.

