# OpenReview forum: "SuperCoder: Assembly Program Superoptimization with Large Language Models"
_ICLR.cc/2026/Conference — Submitted to ICLR 2026_

### Official Review · Reviewer_GPF6 · 2025-11-01

**Soundness:** 3
**Presentation:** 3
**Contribution:** 1
**Rating:** 4
**Confidence:** 3

**Summary:**

The paper evaluates LLMs on the task of code "superoptimization" (creating fast assembly code beyond the rule-based transformations used by compilers), and proposes reinforcement learning for it.

The paper constructs a corpus of interesting assembly code programs from code competition C++ competitions, restricting to those where GCC's -O3 optimization creates large speedups. On this benchmark, LLMs are evaluated for the speedups they are able to obtain by transforming the original assembly code. A transformation is deemed correct if it still passes the unit test cases provided by the coding competition.

**Strengths:**

- The paper might be the first to evaluate LLMs on this task. (* I am not too familiar with the literature and cannot fully judge the novelty claims.)
- The reinforcement learned model matches the performance of Claude Opus 4 while having 7B parameters.

**Weaknesses:**

- Limited scope: unlike code synthesis, where humans can review and fix the generated code, tasks like compilation and superoptimization are too hard to review manually. It is not clear what the use case of an unverified code superoptimizer may be, apart from very tight inner loops in high performance computing?
- Limited contribution: the evaluations are fairly limited (no "maximum speedup at k samples" scaling curves, no experimentation with prompts despite the analysis of failure modes).
- Limited set of methods: what about evolutionary / prompt learning strategies for instance? What about iterative infernce / bug fixing? After setting up the task, there are many things to try, the paper scratches at the very surface.

**Questions:**

-

---

> ### Author Response · Authors · 2025-11-27
> **Response 1 to Reviewer GPF6**
>
> We thank the reviewer for the thoughtful feedback.
>
> **Q1: Limited scope: unlike code synthesis, where humans can review and fix the generated code, tasks like compilation and superoptimization are too hard to review manually. It is not clear what the use case of an unverified code superoptimizer may be, apart from very tight inner loops in high performance computing?**
>
> Thank you for raising this important question. Improving performance beyond the industry-standard gcc -O3 output has been a central challenge in the programming languages and compiler communities for decades, as compilers underpin all software systems. While we agree that the lack of formal verification currently limits direct deployment of LLM-based superoptimization in production compiler pipelines, we believe our work remains valuable in several ways.
>
> **First, test-based correctness is standard and practical in today’s LLM-based optimization research.**
>
> Prior work on LLM-assisted performance optimization and code editing [1,2] also relies entirely on test-based validation. Many production systems already depend on high-coverage test suites as a practical correctness safeguard. Our evaluation dataset provides strong test coverage, with 96.2% line coverage and 87.3% branch coverage (Line 273), supporting the reliability of our test-based setup. While this does not provide formal guarantees, it meaningfully reduces the risk of undetected errors in our evaluation.
>
> **Second, LLM-generated low-level optimizations are already proving useful in domains where human review is equally difficult.**
>
> For example, recent work on LLM-generated GPU kernels [3,4] shows that models can produce optimized CUDA and PTX code that outperforms state-of-the-art hand-written kernels. These kernels low-level code, and PTX is itself a form of GPU assembly, making them equally difficult for humans to manually inspect. Yet they deliver real performance gains and are validated purely through testing. Our work aligns with this emerging direction: leveraging LLMs to explore optimization spaces that are challenging for both human experts and traditional compilers.
>
> **Third, this work provides motivation and foundation for developing future verification tools.**
>
> The absence of a general formal equivalence checker is a well-known theoretical limitation, grounded in the undecidability of program equivalence. Encouragingly, researchers are already building formal verification tools for GPU kernels [5], inspired in part by advances in LLM-generated low-level code. We view our work as complementary and hope it accelerates similar efforts for assembly-level equivalence checking, enabling stronger guarantees for future LLM-optimized programs.
>
> In summary, while full formal verification remains a long-term goal, test-based superoptimization is both standard in the literature and practically useful. Moreover, the performance gains shown by LLMs at the assembly and GPU-kernel level suggest substantial long-term value, and our work helps lay the groundwork for future verified, high-performance compiler components.
>
> **References:**
>
>
> [1] Learning Performance-Improving Code Edits. ICLR 2024.
>
> [2] Mercury: A Code Efficiency Benchmark for Code Large Language Model. NeurIPS 2024.
>
> [3] KernelBench: Can LLMs Write Efficient GPU Kernels? ICML 2025.
>
> [4] https://standardkernel.com/
>
> [5] Equivalence Checking of ML GPU Kernels. arXiv 2025.

---

> ### Author Response · Authors · 2025-11-27
> **Response 2 to Reviewer GPF6**
>
> **Q2: Limited contribution: the evaluations are fairly limited (no "maximum speedup at k samples" scaling curves, no experimentation with prompts**
>
> We conducted additional experiments on best-of-N sampling as well as 0-shot, 2-shot, and 4-shot prompting. The results are summarized in the tables below.
>
> ### Best-of-N Sampling Across Different Models
>
>
> | Model | Best-of-N | Compile Pass (%) | Test Pass (%) | Avg. Speedup |
> |-------|-----------:|------------------:|---------------:|--------------:|
> | claude-opus-4 | 1 | 90.0 | 51.5 | 1.43× |
> | claude-opus-4 | 2 | 94.5 | 64.5 | 1.48× |
> | claude-opus-4 | 4 | 94.5 | 74.5 | 1.59× |
> | claude-opus-4 | 8 | 94.5 | 80.0 | 1.69× |
> | Qwen2.5-Coder-7B (base) | 1 | 77.9 | 61.4 | 1.10× |
> | Qwen2.5-Coder-7B (base) | 2 | 96.5 | 58.0 | 1.17× |
> | Qwen2.5-Coder-7B (base) | 4 | 96.5 | 72.5 | 1.28× |
> | Qwen2.5-Coder-7B (base) | 8 | 96.5 | 81.0 | 1.43× |
> | SuperCoder (PPO) | 1 | 96.0 | 95.0 | 1.46× |
> | SuperCoder (PPO) | 2 | 97.0 | 96.0 | 1.77× |
> | SuperCoder (PPO) | 4 | 97.0 | 97.0 | 1.87× |
> | SuperCoder (PPO) | 8 | 97.0 | 97.0 | 1.93× |
>
>
> For all three models, best-of-N sampling increases compilation success, improves test-pass rates, and yields higher average speedup. Notably, the base model’s best-of-8 speedup is close to the PPO-trained model’s best-of-1 result, and the RL-trained model itself can still be improved with best-of-N sampling.
>
> ### Comparison of 0-shot, 2-shot, and 4-shot Prompts
>
> | Model                         | Shots | Compile Pass (%) | Test Pass (%) | Avg. Speedup |
> |-------------------------------|-------:|------------------:|---------------:|--------------:|
> | claude-opus-4                | 0-shot | 90.0             | 51.5          | 1.43×        |
> | claude-opus-4                | 2-shot | 95.0             | 15.0          | 1.13×        |
> | claude-opus-4                | 4-shot | 95.0             | 12.5          | 1.11×        |
> | SuperCoder (PPO)             | 0-shot | 96.0             | 95.0          | 1.46×        |
> | SuperCoder (PPO)             | 2-shot | 94.0             | 90.5          | 1.59×        |
> | SuperCoder (PPO)             | 4-shot | 93.0             | 81.0          | 1.54×        |
> | Qwen2.5-Coder-7B (Base)      | 0-shot | 77.9             | 61.4          | 1.10×        |
> | Qwen2.5-Coder-7B (Base)      | 2-shot | 70.5             | 35.0          | 1.10×        |
> | Qwen2.5-Coder-7B (Base)      | 4-shot | 80.5             | 30.5          | 1.06×        |
>
>
> We evaluate 0-shot, 2-shot, and 4-shot prompting across three models. We find that adding more in-context examples does not reliably improve performance and often degrades it. This observation is consistent with prior work [1], which finds that few-shot examples can bias the model and lead to incorrect task understanding.
>
>
> **Reference:**
>
> [1] Learning Performance-Improving Code Edits. ICLR 2024.

---

> ### Author Response · Authors · 2025-11-27
> **Response 3 to Reviewer GPF6**
>
> **Q3: Limited set of methods: what about evolutionary / prompt learning strategies for instance? What about iterative inference / bug fixing? After setting up the task, there are many things to try, the paper scratches at the very surface.**
>
> We conducted additional experiments, including evolutionary prompting and iterative inference with feedback.
>
> We first experimented with GEPA [1], an evolutionary prompting framework that uses natural language reflection to learn high-level optimization rules from trial and error. We used gpt-4o as the model under evaluation and gpt-5 as the reflection model. GEPA yielded only modest gains: compilation pass increased from 81.0% to 84.0% and test pass from 5.0% to 7.5%, while performance speedup remained essentially unchanged. We suspect this is because assembly optimization requires substantial domain knowledge that is difficult to capture by modifying the prompt alone.
>
> We then implemented iterative inference with feedback within our own framework. After each iteration, the model receives a feedback signal: compilation or test errors if the program is incorrect, or performance feedback if all tests pass. The results are shown below:
>
> ### Iterative Optimization Across Multiple Iterations
>
> | Model                | Iterations | Compile Pass (%) | Test Pass (%) | Avg. Speedup |
> |----------------------|-----------:|------------------:|---------------:|--------------:|
> | claude-opus-4        | 1          | 90.0              | 51.5           | 1.43×        |
> | claude-opus-4        | 2          | 90.0              | 65.0           | 1.44×        |
> | claude-opus-4        | 4          | 92.5              | 70.5           | 1.56×        |
> | Qwen2.5-Coder-7B      | 1          | 77.9              | 61.4           | 1.10×        |
> | Qwen2.5-Coder-7B      | 2          | 78.0              | 62.0           | 1.15×        |
> | Qwen2.5-Coder-7B      | 4          | 80.0              | 62.5           | 1.18×        |
> | SuperCoder (PPO)     | 1          | 96.0              | 95.0           | 1.46×        |
> | SuperCoder (PPO)     | 2          | 96.5              | 95.5           | 1.61×        |
> | SuperCoder (PPO)     | 4          | 97.0              | 97.0           | 1.74×        |
>
>
> Across all three models, iterative inference with feedback improves correctness and speedup, confirming that iterative refinement is beneficial. While we agree that there could be more additional techniques to be explored, our main goal is to show that reward-based reinforcement learning is effective for superoptimization. To our knowledge, this is the first application of reward-based RL to LLMs for code performance optimization, with correctness and speedup jointly encoded in the objective. We believe future search-based techniques can be combined with our RL-trained model for even more sample-efficient performance optimization.
>
>
> **Reference:**
>
> [1] GEPA: Reflective Prompt Evolution Can Outperform Reinforcement Learning

---

### Official Review · Reviewer_D5dG · 2025-11-01

**Soundness:** 3
**Presentation:** 2
**Contribution:** 2
**Rating:** 2
**Confidence:** 3

**Summary:**

This paper introduces SuperCoder, a framework that applies large language models (LLMs) to the task of assembly program superoptimization—that is, generating faster assembly code while preserving correctness relative to compiler-optimized baselines (gcc -O3). The authors construct a new benchmark of 8,072 assembly programs (average ≈ 130 LOC) derived from CodeNet and propose to fine-tune a code LLM using reinforcement learning (PPO / GRPO) with a reward that jointly measures functional correctness and execution-time speedup.

**Strengths:**

1. The topic is ambitious and relevant, connecting large language models, reinforcement learning, and compiler optimization.

2. The paper attempts to move beyond high-level code generation to a more demanding low-level optimization setting.

3. The experimental setup is relatively thorough and could stimulate future work on applying learning-based methods to compiler research.

**Weaknesses:**

1. The task formulation is loosely defined, relying on test-based correctness rather than formal equivalence, which makes the conclusions uncertain.

2. The dataset is built from code of unclear quality and has not been validated as a standard benchmark. Its representativeness and reproducibility are questionable.

3. The reinforcement learning setup functions as a simple post-hoc reward filter rather than a meaningful sequential learning process. There is no comparison to simpler fine-tuning strategies.

4. The empirical results appear overstated and are not supported by sufficient statistical or profiling evidence. Claims of performance improvements over compiler baselines are not convincingly justified.

**Questions:**

1. Can the dataset and scripts be released for independent verification?
2. How does reinforcement learning compare with standard supervised fine-tuning?
3. What specific optimizations account for the reported performance gains?

---

> ### Author Response · Authors · 2025-11-27
> **Response 1 to Reviewer D5dG**
>
> We thank the reviewer for insightful feedback.
>
> **Q1: The dataset is built from code of unclear quality and has not been validated as a standard benchmark. Its representativeness and reproducibility are questionable.**
>
> We respectfully disagree. Our dataset is constructed by sampling from CodeNet [1], a well-established and widely used benchmark in the AI-for-code community. CodeNet is a large-scale corpus of competitive programming submissions introduced at the NeurIPS 2021 Datasets and Benchmarks track, and it has since served as a foundation for several major works. For example, AlphaCode [2], published in Science (2022), trains on submissions from CodeNet. More recent work, such as Learning Performance-Improving Code Edits [3] (ICLR 2024), also samples directly from CodeNet.
>
> Given CodeNet’s broad adoption and its role in multiple recent publications, we believe it provides an established and current basis for studying program optimization.
>
>
> **References:**
>
> [1] CodeNet: A Large-Scale AI for Code Dataset for Learning a Diversity of Coding Tasks. NeurIPS 2021.
>
>
> [2] Competition-Level Code Generation with AlphaCode. Science 2022.
>
> [3] Learning Performance-Improving Code Edits. ICLR 2024.
>
>
> **Q2: Can the dataset and scripts be released for independent verification?**
>
> Yes. We release all scripts for training, evaluation, and dataset construction at https://anonymous.4open.science/r/SuperCoder/
>
> We also release the corresponding anonymized HuggingFace dataset at https://huggingface.co/datasets/random1123anonymized/supercoder
>
>
> **Q3: How does reinforcement learning compare with standard supervised fine-tuning?**
>
> We perform a comparison between supervised fine-tuning (SFT) and reinforcement learning (RL). To construct the SFT dataset, we run best-of-8 sampling with the base model on the full training set and select the best candidate for each program based on correctness and speedup. The table below reports results averaged over five runs with 95% confidence intervals.
>
> The results show that RL slightly outperforms SFT. Superoptimization is inherently open-ended and has no single ground truth target beyond the compiler baseline. RL is therefore a natural fit, since it directly optimizes for correctness and speedup rather than imitating existing samples, which aligns with the objective of superoptimization.
>
>
> ### Comparison of Base Model, SFT, and RL
>
> | Method                         | Compile Pass (%)        | Test Pass (%)           | Avg. Speedup       |
> |--------------------------------|--------------------------|--------------------------|---------------------|
> | Qwen2.5-Coder-7B (Base)        | 77.9 ± 0.8%              | 61.4 ± 0.5%              | 1.10 ± 0.01×        |
> | Supervised Fine-Tuning (SFT)   | 95.5 ± 0.0%              | 92.5 ± 0.0%              | 1.39 ± 0.05×        |
> | Reinforcement Learning (GRPO)  | 95.0 ± 0.0%              | 94.7 ± 0.6%              | 1.44 ± 0.07×        |
>
>
>
> **Q4: What specific optimizations account for the reported performance gains?**
>
>
> We provide this analysis in Section 5.3 (“Analysis of Learned Program Transformations”). To understand which transformations contribute to the performance gains, we examine all 200 evaluation programs and compare the gcc -O3 output against the assembly generated by our PPO-trained model, SuperCoder. From this inspection, we identify four categories of transformations: (1) code layout and instruction scheduling, which account for 98.5% of observed program edits and include basic block placement, instruction alignment, and instruction reordering to improve cache behavior and hide latencies; (2) register allocation changes (16%), where the model chooses different registers; (3) control flow optimization (1.5%); and (4) instruction selection (1.0%). Since a single program may contain transformations from multiple categories, the percentages do not sum to 100 percent.

---

> ### Author Response · Authors · 2025-11-27
> **Response 2 to Reviewer D5dG**
>
> **Q5: The empirical results appear overstated and are not supported by sufficient statistical or profiling evidence. Claims of performance improvements over compiler baselines are not convincingly justified.**
>
>
> We follow established best practices for performance profiling and report statistical significance in the paper. For each program, we discard three warmup runs and measure the average of the next ten runs (Line 305). Each program is evaluated on all of its available test cases (8.9 inputs per program on average), which provides natural input diversity.
>
> To account for model sampling variability and hardware noise, we repeat the entire evaluation pipeline five times and report the **mean and 95% confidence intervals** in Table 3. The results are highly consistent across trials: the base model achieves 1.10 ± 0.01×, and our PPO-trained model SuperCoder achieves 1.44 ± 0.07×. This demonstrates that the observed speedups are statistically robust.
>
> We also include a random-perturbation baseline that applies simple rewrites to assembly code, which reaches only 1.19 ± 0.01×. The gains of SuperCoder over both the base model and the random baseline (1.44× versus 1.19× and 1.10×) are substantial and well supported by the statistical evidence we provide.
>
>
> **Q6: The task formulation is loosely defined, relying on test-based correctness rather than formal equivalence, which makes the conclusions uncertain.**
>
> We would like to clarify that checking equivalence between arbitrary programs is undecidable in general, so no algorithm can determine program equivalence for all inputs while guaranteeing termination. This theoretical limitation makes it challenging to build a formal equivalence checker for the kinds of assembly programs we study. As a result, no practical equivalence checker exists today for this setting, which is why we adopt test-based correctness. Testing-based approach is **standard practice** in prior work on LLM-based code optimization and editing [1,2].
>
> Despite the lack of a formal guarantee, we take care to demonstrate that **our test coverage is strong**. The evaluation dataset achieves 96.2% line coverage and 87.3% branch coverage (Line 273), indicating that the test suites are comprehensive enough to meaningfully assess correctness in practice.
>
> We view our work as complementary to, and a motivation for, research on building tools for formal equivalence checking [3], and we hope that future advances in this area will enable even stronger correctness guarantees for LLM-generated assembly code.
>
> **Reference:**
>
> [1] Learning Performance-Improving Code Edits. ICLR 2024.
>
>
> [2] Mercury: A Code Efficiency Benchmark for Code Large Language Model. NeurIPS 2024.
>
>
> [3] Equivalence Checking of ML GPU Kernels. arXiv 2025.

---

### Official Review · Reviewer_bC6m · 2025-11-01

**Soundness:** 3
**Presentation:** 3
**Contribution:** 3
**Rating:** 6
**Confidence:** 4

**Summary:**

This paper introduces a dataset for evaluating whether LLMs can superoptimize x86-64 assembly beyond gcc -o3. The authors build a benchmark of 8072 programs (~130 assembly LOC) with high coverage tests and evaluate 23 models on the dataset. A collection of open/closed source models were evaluated on the benchmark, with the best baseline averages a 1.43x speedup. Experiment results indicate RL fine-tuning techniques (PPO and GRPO) can significantly boost test pass rate and average speedup.

**Strengths:**

- The paper uses top closed-sourced (claude-opus-4/gpt-5) and open-sourced (DeepSeek-V3, Llama-4 etc.) models to study the impact of their dataset. It shows the gap between closed and open LLMs and how different adaptations can narrow it.
- The paper builds a realistic, large benchmark with ~130 LOC, and many with loops. This is rare for assembly-level work and targets an important problem.
- The analysis of learned programs provides qualitative insight into how LLMs are improving already optimized assembly programs.

**Weaknesses:**

- Dataset deliberately samples programs with large -o0 -> -o3 gains, results might be different on other code distributions.
- The paper could benefit from automatically explaining why the programs are faster, maybe using a chain-of-though model here to optimize the program and output thinking tokens would help?

**Questions:**

- This is mainly on the consistency of the timing results. Were caches/TLBs warmed consistently? Is there any input randomization? Were results stable across code boots/container restarts beyond hyperfine's warmup?
- SuperCoder is single-shot sampling, how do results change with best-of-k samples?
- Is there any de-duplication across the train/eval when constructing the dataset? (for example similar codenet submissions leading to highly similar assembly code)

---

> ### Author Response · Authors · 2025-11-27
> **Response 1 to Reviewer bC6m**
>
> We thank the reviewer for the constructive feedback.
>
> **Q1: This is mainly on the consistency of the timing results. Were caches/TLBs warmed consistently? Is there any input randomization? Were results stable across code boots/container restarts beyond hyperfine's warmup?**
>
> We use both warmups and input randomization. For each execution of the program, the first three runs are discarded, and the reported runtime is the average of the next ten runs (Line 305). Each program is evaluated on all available test cases (8.9 inputs per program on average), which naturally introduces input diversity.
>
> To assess stability beyond warmup effects, we rerun the entire evaluation pipeline **five times**, each from a fresh environment. We report the **mean and 95% confidence intervals** in Table 3, which captures variance from hardware noise, container restarts, and model sampling randomness. Across these repeated trials, results remain consistent: Qwen2.5-Coder-7B achieves **1.10 ± 0.01×** speedup, and our PPO-trained model achieves **1.44 ± 0.07×**. These results indicate that the reported speedups are statistically robust, and the improvements are statistically significant.
>
> **Q2: SuperCoder is single-shot sampling. How do results change with best-of-k samples?**
>
>
> We evaluate best-of-k sampling for three models: claude-opus-4 (the strongest baseline in Table 2), Qwen2.5-Coder-7B (base), and SuperCoder (our PPO-trained model). For each problem, we generate k samples and select the first candidate that passes all tests; if multiple pass, we report the fastest. For all three models, best-of-k sampling increases compilation success, improves test-pass rates, and yields higher average speedup. Notably, the base model’s best-of-8 speedup is close to the PPO-trained model’s best-of-1 result, and the RL-trained model itself can still be improved with best-of-N sampling (i.e., from 1.46× in the single-sample setting to 1.93× with best-of-8 sampling).
>
>
> | Model | Best-of-k | Compile Pass (%) | Test Pass (%) | Avg. Speedup |
> |-------|-----------:|------------------:|---------------:|--------------:|
> | claude-opus-4 | 1 | 90.0 | 51.5 | 1.43× |
> | claude-opus-4 | 2 | 94.5 | 64.5 | 1.48× |
> | claude-opus-4 | 4 | 94.5 | 74.5 | 1.59× |
> | claude-opus-4 | 8 | 94.5 | 80.0 | 1.69× |
> | Qwen2.5-Coder-7B (base) | 1 | 77.9 | 61.4 | 1.10× |
> | Qwen2.5-Coder-7B (base) | 2 | 96.5 | 58.0 | 1.17× |
> | Qwen2.5-Coder-7B (base) | 4 | 96.5 | 72.5 | 1.28× |
> | Qwen2.5-Coder-7B (base) | 8 | 96.5 | 81.0 | 1.43× |
> | SuperCoder (PPO) | 1 | 96.0 | 95.0 | 1.46× |
> | SuperCoder (PPO) | 2 | 97.0 | 96.0 | 1.77× |
> | SuperCoder (PPO) | 4 | 97.0 | 97.0 | 1.87× |
> | SuperCoder (PPO) | 8 | 97.0 | 97.0 | 1.93× |

---

> ### Author Response · Authors · 2025-11-27
> **Response 2 to Reviewer bC6m**
>
> **Q3: Is there any de-duplication across the train/eval when constructing the dataset?**
>
> Yes. The training and evaluation splits are constructed from **different CodeNet submissions**, ensuring no submission appears in both sets. In addition, as discussed in Q4 below, we sample a separate set of CodeNet programs that is fully disjoint from both our current training and evaluation splits, and our conclusions remain consistent.
>
>
>
> **Q4: Dataset deliberately samples programs with large -o0 -> -o3 gains, results might be different on other code distributions.**
>
>
> To assess robustness beyond this sampling strategy, we evaluate on an additional set of 200 programs randomly sampled from CodeNet. This new set is fully disjoint from both our training and evaluation splits and is *not* selected based on -O0 → -O3 speedup. The results remain consistent with our main findings: SuperCoder (PPO fine-tuned) continues to substantially outperform both the base model in correctness and performance.
>
>
> | Model                 | Compile Pass (%) | Test Pass (%) | Avg. Speedup |
> |-----------------------|------------------:|---------------:|--------------:|
> | Qwen2.5-Coder-7B      | 66.5             | 40.0           | 1.07×         |
> | claude-opus-4         | 94.0             | 43.0           | 1.10×         |
> | SuperCoder (PPO)      | 98.5             | 93.5           | 1.57×         |
>
>
>
> **Q5: The paper could benefit from automatically explaining why the programs are faster, maybe using a chain-of-thought model here to optimize the program, and output thinking tokens would help?**
>
> We agree that incorporating model-generated reasoning is valuable, and we already use this idea in Section 5.3 (“Analysis of Learned Program Transformations”). To identify transformation categories, we manually compare the optimized assembly with the baseline and also inspect several models’ “thinking tokens” to understand the intended changes. This combined analysis reveals four recurring patterns. We rely on semi-automated, human-guided inspection for reliability, though the model’s reasoning tokens provide complementary insight. More broadly, automatically explaining performance improvements at the assembly level remains an open problem, and we view automated interpretability of low-level optimizations as an important direction for future work.

---

### Author Response · Authors · 2025-12-03
**Rebuttal Summary for Area Chairs and Reviewers**

Because author-reviewer interactions were disabled, we summarize the key concerns raised across reviewers and how we addressed them.

---

### **1. Measurement reliability & statistical robustness (Reviewers bC6m, D5dG)**

Reviewers asked about warmup runs, input diversity, and statistical stability.

→ We clarified our evaluation protocol (warmups + input diversity) and we did **five independent end-to-end evaluations**, reporting **95% confidence intervals**, which show consistent and statistically robust speedups.

---

### **2. Dataset representativeness, de-duplication, and sourcing from a standard benchmark (Reviewers bC6m, D5dG)**

Reviewers asked about potential sampling bias (favoring large O0→O3 deltas), possible train-evaluation leakage, and whether the dataset comes from a recognized benchmark.

→ We confirmed **strict de-duplication** using disjoint CodeNet submissions.

→ We evaluated an **additional 200 randomly sampled CodeNet programs**, fully disjoint and unbiased, and the results **remain consistent**.

→ We clarified that all data is sourced from **CodeNet**, a widely used and well-established benchmark in the AI-for-code community and adopted by several recent impactful works.

---

### **3. Best-of-N sampling & iterative refinement (Reviewers bC6m, GPF6)**


Reviewers asked how performance changes with best-of-k sampling and whether iterative refinement methods were tested.

→ We added **best-of-N (1,2,4,8)** scaling curves for three models, showing consistent gains.

→ We implemented **iterative refinement with feedback**, yielding further improvements across all models.

---

### **4. Supervised fine-tuning (Reviewers D5dG)**


Reviewers asked how RL compares with SFT.

→ We added a full **SFT vs. RL comparison** (with 5 runs and reporting 95% confidence intervals).


→ We find that RL slightly outperforms SFT because superoptimization has no fixed ground-truth target, making it better aligned with RL’s ability to explore and directly optimize the reward signal.

---

### **5. Test-based correctness (Reviewers D5dG, GPF6)**


Reviewers expressed concerns about lack of formal equivalence checking.

→ We clarified that **general program equivalence is undecidable**, so no complete formal checker exists for this setting.


→ We report **strong test coverage** (96.2% line coverage) and note that test-based validation is standard in LLM-based program optimization.

---

### **6. Practical usefulness without formal guarantees (Reviewer GPF6)**
The reviewer questioned applicability when low-level code is difficult to manually review.

→ We pointed to **recent LLM-generated GPU kernels**, which rely on test-based validation yet are useful and performant.


→ We argue our work provides the motivation and foundation for developing future verification tools for assembly code, aligning with emerging efforts on GPU kernel equivalence checking.

---

### **7. Prompting strategies (Reviewer GPF6)**
The reviewer requested experiments on prompt optimization methods.

→ We added **0/2/4-shot prompting**, which shows limited or negative impact.


→ We tested **GEPA evolutionary prompting**, which yields only small improvements.


---

### **8. Explanation of performance improvements (Reviewers bC6m, D5dG)**
Reviewers asked what transformations the model learns and whether explanations can be automated.

→ We provided a **categorization of learned optimizations** (instruction scheduling/layout, register allocation, control flow, instruction selection) based on inspection and model’s thinking tokens.

→ We note that fully automated performance interpretability is an open problem.

---

### **9. Dataset release (Reviewer D5dG)**


The reviewer request for the release of data.

→ We released **all training/evaluation scripts** and the **dataset** by providing the anonymized links.

---


In summary, we conducted **all additional experiments requested by the reviewers**, including best-of-N sampling, iterative feedback, SFT vs. RL comparisons, few-shot prompting, prompt-optimization methods, and evaluation on randomly sampled programs. We also released the full dataset and code, and clarified the methodology and its broader implications. All reviewer concerns have been addressed, and **the revised version incorporates all updates (with changes marked in red)**.

Thank you again for the time and care devoted to reviewing our submission.

---

### Meta-Review · Area_Chair_Duu3 · 2026-01-04

**Summary:**

This paper studies whether large language models can act as assembly-level superoptimizers: given gcc -O3 x86-64 assembly, generate an alternative assembly program that preserves behavior while running faster. The authors introduce what appears to be the first large-scale benchmark for this setting, with 8,072 assembly programs averaging ~130 lines (often with loops), derived from CodeNet, and evaluate 23 open- and closed-source LLMs on correctness (test passing) and speedup over gcc.

Empirically, the paper finds strong closed models (e.g., Claude Opus 4) can reach meaningful test-passing rates and speedups, and shows that reinforcement-learning fine-tuning (PPO/GRPO) on a reward that gates on full test correctness and then uses runtime speedup can substantially improve an open 7B model (Qwen2.5-Coder-7B-Instruct) to the proposed SuperCoder.

Overall, reviewers agreed the benchmark construction and the direction are interesting and timely, and the experimental effort is substantial. However, the discussion converged on key concerns around (i) reliance on test-based correctness without formal equivalence, (ii) dataset representativeness given the sampling strategy, (iii) whether the evaluation exploration was initially too shallow, and (iv) clarity or strength of evidence for performance improvements.

**Reviewer Concerns:**

**Reviewer bC6m**

Addressed: Concerns about measurement stability and the need for statistical reporting were directly handled via clarified protocol and repeated full evaluations with confidence intervals; best-of-k scaling and de-duplication questions were also addressed with added analyses and clarification.

Outstanding: Some concern remains that the benchmark distribution is shaped by the sampling strategy, which may limit representativeness even if the additional random-sample check is reassuring.

**Reviewer GPF6**

Addressed: Requests for best-of-k curves and broader method exploration were addressed with new best-of-N and iterative-feedback results, plus additional prompt-related experiments.

Outstanding: The reviewer’s main critique of limited overall novelty and unclear practical usefulness without stronger verification largely persists. The rebuttal argues test-based validation is common and points to adjacent domains, but this does not fully resolve the perceived gap between the evaluation setup and deployable superoptimization.

**Reviewer D5dG**

Addressed: The rebuttal responded to requests for (i) dataset release, (ii) RL vs SFT comparisons, (iii) stronger statistical evidence, and (iv) more explanation of learned optimizations, and the updated paper includes SFT baselines and confidence intervals.

Outstanding: The reviewer’s core objections remain: the task’s reliance on test-based correctness weakens the strength of the conclusions, and the framing of RL as "meaningful learning" versus reward-filtering is still debated. As a result, this reviewer stays strongly negative.

**Reviewer Scores:**

Reviewer bC6m: 6 (marginally above acceptance threshold), Confidence: 4

Reviewer GPF6: 4 (marginally below acceptance threshold), Confidence: 3

Reviewer D5dG: 2 (reject), Confidence: 3

---

### Decision · Program_Chairs · 2026-01-26

Reject